# The STEM Crisis and Teacher Practice: Exploring Responses to the Competing Discursive Arrangements of Education in the Sciences in a Catholic School Setting

Simon N. Leonard [1,*], Lisa O'Keeffe [1], Bruce White [1], Melanie O'Leary [2] and Karen Sloan [3]

1   Education Futures, University of South Australia, Adelaide 5000, Australia
2   Catholic Education South Australia, Adelaide 5000, Australia
3   School of Education, University of Adelaide, Adelaide 5005, Australia
*   Correspondence: simon.leonard@unisa.edu.au

**Abstract:** STEM has become a pervasive part of global education reform. The STEM discourse positions the purpose of scientific education as being to prepare young people for work in a hyper-competitive 21st century knowledge economy, pushing aside alternative approaches focussed on interrogating social, moral and political issues in context. This narrative does not always sit comfortably with the holistic ambitions of many state and faith-based education systems. In this paper we will argue that these tensions emerge from deeper conflicts in the cultural-discursive arrangements around education in the advanced democratic states through an exploration of the response to a STEM curriculum project in a Catholic education system. The exploration is based on a phenomenographic analysis of reflective interviews conducted with participating teachers. We conclude that while the teachers are aware of the tensions, they may benefit from access to a language for discussing the various pressures on learning design and meaning making.

**Keywords:** STEM practices; socio-scientific decision making; practice theory; practice architectures; STEM; Catholic education

## 1. Introduction

### 1.1. Architectures of Practice

Our purpose in this paper is to explore how teachers interact with the 'architectures' of their practices [1] when those architectures create a contested practice space. The idea of practice architectures is drawn from Kemmis' development of practice theory, in which he argues that practices occur within semiotic, physical and social space, and so are shaped by the cultural-discursive, material-economic and socio-political architectures that surround it [1].

In this paper we are particularly interested in the discursive architectures being experienced by teachers during a multi-year 'STEM' curriculum project being conducted by a large Catholic school system in Australia. We will access these arrangements through a methodology known as computer aided phenomenography [2,3]. Our source data for analysis are the reflective accounts of practice provided by teachers in interviews taken during the project. Through this analysis, and by borrowing the concept of 'teacher democratic assignment' from Mooney Simmie and Edling [4] we will argue that the teachers involved in the project were in need of a more coherent discursive practice architecture to support innovative the innovations to practice that the project sought.

Kemmis' architectures framework [1,5] builds on theorisations of practice offered by Schatzki [6], Wittgenstein [7] and MacIntyre [8] to offer a definition of practice as:

> . . . *a form of socially established cooperative human activity in which characteristic arrangements of actions and activities (doings) are comprehensible in terms of arrangements of relevant ideas in characteristic discourses (sayings), and when the people and*

*objects involved are distributed in characteristic arrangements of relationships (relatings), and when this complex of sayings, doings and relatings 'hangs together' in a distinctive project.* [9] (p. 31)

Flowing from this definition, the 'practice architectures' and 'practice traditions' of Kemmis are similar in some ways to Schatzki's conception of 'practice memory'. It is built on an understanding that social actions are located not simply in the minds of the individual participants but also in the shared language, materiality and social structures that surround people and their practices. That is, practice is 'stored' not only in our minds, but the way we speak, the tools we use and way we interact with each other. More formally, Kemmis's approach is interested in the cultural-discursive arrangements that occur in semantic space in the medium of language, in the material-economic arrangements that occur in physical space-time in the medium of activity or work, and in the social-political arrangements that occur in social space in the medium of solidarity or power [9]. Further, it is interested in the interactions of these arrangements that simultaneously shape, and are shaped by, practice.

In this paper we will focus on an analysis of the discursive arrangements reflected by teachers in their recounts of their involvement in the STEM curriculum project. It is worth noting that our purpose and approach are distinctly different to phenomeno*logical* studies seeking to understand teacher and/or student experiences of curriculum enactment. Rather, through our phenomeno*graphic* approach we are seeking to map the range of experiences of the present cohort. We are then seeking to use that map to shine an interpretative light upon the architectures surrounding that collective experience [10].

### 1.2. STEM as a Policy Architecture

STEM is notionally an acronym for science, technology, engineering and mathematics although—importantly for this study—it is also a signifier of a very particular approach to and purpose for learning in those subjects. While STEM does promote research and practice at the intersections of these related disciplines [11], we will argue that the teachers in the project experienced a real tension between the discursive arrangements found around 'STEM'—an agenda which configures students as 'accountable subjects' within a preconfigured global future [12]—and the alternative philosophies on how we might respond to the growing list of socio-scientific challenges facing our society.

The alternatives were prominent in this project in a Catholic school system, whose overarching policy frameworks reflected the Vatican's Encyclical Letter Laudato si' [13] that publicly called for the development of young people around core values like sustainability, interconnectedness and dignity. Our paper, though, should be read as having relevance well beyond the Catholic context as that church clearly is not the only source promoting concepts like sustainability and interconnectedness. Indeed, despite their greatly different organisations goals and imperatives, the Organisation for Economic Co-operation and Development (OECD) draw on very similar concepts in promoting 'transformative competencies' like 'taking responsibility' and 'reconciling tensions and dilemmas' [14] as desirable outcomes for children's education.

We will go on to argue that future professional learning must respond to these tensions and assist teachers to establish and re-establish practices of teaching and learning in the science that better 'hang' together with the architectures that support them.

### 1.3. An 'Architecture' for This Research

In exploring the architectures of teaching practice in the sciences, this paper necessarily asks questions of the kind of science education that is desirable more broadly. This is a socio-politically vexed and contested question, and a space in which we as qualitative researchers have a duty to position ourselves within to assist our readers to understand the influences on the analyses we make in a paper such as this. As the lead author, I seek to do this by sharing with you, our reader, a story of the Australian context in which we engaged with this project and the subsequent analysis.

Our context was one of continental climate catastrophe. It was, and remains, a context in which questions on the purpose of science education have perhaps never seemed so visceral. In another era the threat of nuclear war brought with it a similar existential anxiety perhaps, but even that was relatively abstract. In Australia in recent years, however, the failures of our socio-scientific decision making [15] have terrorised us and surrounded us. Through seemingly endless bush (wild) fire we have felt the heat and the soot on our skin, heard the emergency sirens and the helicopters dropping water. We have smelt the smoke that has invaded our homes.

And now the fires have been replaced by floods. Our rivers have risen higher than ever recorded. So called 1 in 100-year flood events are now occurring every few months. As I first looked at the data for this paper though, my country was burning.

Thirty years earlier, my lecturer in physical chemistry told me it would. To a rather small group of undergraduates timetabled for a lecture far too early on a Monday morning he recounted, perhaps with too much joy, the tales of his research into the methane emissions of a waste landfill site. In the Australian vernacular, the 'tip', or the 'dump'. A little less poetically he went on to explain how the electron arrangements of methane could absorb electromagnetic radiation and then spit it back out in the infrared spectrum. Heat. The heat, we learned, would come out in all directions meaning that methane in the atmosphere would send half of heat back towards the planet. The science of this interaction was not overly complex.

Now the places of my undergraduate youth have burnt. The first peoples of this continent tell us that fire has been here since the world was 'dreamed' into existence. But this has been fire unimaginable both to the first peoples, and to the people who have come more recently and named this continent Australia.

It is easy to mythologise, to romanticise, but the summers of this place once were magical. Hot, yes. Fires, yes. And storms and cyclones and droughts and flies. Oh, the flies! But Australia glistened in the summer. Australia had a sky that was larger than the world. Our largest city could reasonably style itself the 'Emerald City' and evoke wonder. Australia was legend.

But as I first read this data there was no sky. That day there was only smoke. Sooty, choking, terrifying smoke. And on the hills the forest, our beautiful 'bush', burnt. And the animals that symbolise our unique place in the world, the kangaroos and koalas and all the others, they died in their millions. Millions! Human life was lost too, as were many more livelihoods. Such devastation, so many tears, so much anger. On the news a phrase is repeated. 'Scientists have been telling us for decades that this would happen'.

Why have we been so unable to listen and to act? And what is the role for science education in the midst of such ongoing destruction?

*1.4. Language-Games*

Our interest in the cultural-discursive arrangements of semiotic space builds on the concept of the language-game offered by Wittgenstein [7]. Language, Wittgenstein argued, was not separate to its corresponding reality. Rather he saw language and activity as interwoven. As parts of each other. In these terms we were interested in the significant tensions in the language-games we saw around the project. Our research question was 'how do the tensions in the language-games of policy impact on teachers' practice in this STEM curriculum project'?

The tensions we were interested in were between a near-global educational policy ensemble that has been labelled 'STEM', and Catholic Education South Australia's learning framework called 'Living Learning Leading' [16]. The Living Learning Leading framework positions schools as a place for dialogue with the world as a people of faith, and sets out the primary purpose of schooling as the development of 'thriving people, capable learners, leaders for the world God desires'. Through their interactions with school, the framework directs, children should be literate and numerate, but also self-aware, spiritually aware, moral, compassionate, interculturally and globally minded, inquisitive and innovative.

While couched in the language of the Catholic Church, the framework reflects a wider philosophical interest in the 'good' human life that can be found in sources as diverse as the OECD—as noted above—and the works of John Dewey [17]. It represents a common cultural-discursive architecture for teacher practice Australia, like many other places.

The narratives of STEM education start from a very different cultural-discursive place. The South Australian STEM Education strategy document, for example, begins with a statement from the State Education Minister that 'We know that 75% of the fastest growing occupations now require STEM skills and knowledge' [18] (p. i). This document commences its argument by asserting that the economic case for STEM is clear, given that 'between 2006 and 2011 in Australia, the number of people in positions requiring STEM qualifications grew 1.5 times faster than all other occupation groups' (p.2). This is a document concerned with human needs almost exclusively as they relate to the needs of the economy.

Critiques of the assumptions of the STEM narrative are now well developed in the literature. Presenting their case in the form of a play for the theatre, for example, Weinstein et al. [19] borrow from Foucault and draw our attention to the way the STEM discourse supports the neoliberal positioning of the market as the centre of a regime of truth. Giving primacy to economic gain within a competitive market, the STEM discourse reflects the ambivalence—and even antipathy—towards expertise that is foundational to neoliberal thinking [20]. Within this regime, what is important, what counts as truth, is that which gives advantage [21]. Within such a regime 'truth' ceases to be deliberative, democratic and expert and is instead focussed on real-time intelligence aligned with an aggressive promotion of superiority. As a result, for example, we see the universities turn their focus to the enforcement of intellectual property over and above the creation and dissemination of knowledge for the public good [22,23].

The language-game of STEM does more than simply align the purpose of the sciences with the economy, it seeks to promote a particular kind of economic reality. STEM is spoken of almost interchangeably with entrepreneurship and constructs workers of the future as 'free, enterprising individuals who govern themselves' [24] requiring only limited direct control from the state. At the same time STEM very much bonds workers to the interests of the state and calls to maintain national positions in a global 'STEM race'. Like the Olympics, STEM quickly starts to sound like war carried out by other means.

Feminist and decolonising theories of science have offered important insights into the language-games of STEM. In that work we find arguments that the extensive focus on increasing the participation of underrepresented groups such as women in STEM educational and professional settings has counterintuitively—and perhaps counterproductively—served to actually entrench hegemonic gendered and racist conceptions of science. Heybach and Pickup [25], for example, argue through an analysis of the production of STEM toys aimed at girls that the strong programmatic focus on inclusion of girls has positioned STEM as a neutral commodity to be distributed rather than as a domain of knowledge impacted by the experiences of gender. Drawing on feminist work from science and technology studies [26] they argue that science itself might be different if conceived and practiced from a feminist perspective.

The critique of the dominant STEM discourse was particularly topical for a project in Catholic schools. Under the leadership of Pope Francis, in recent years the Catholic Church has been reassessing its relationship with expertise and science. This is seen most prominently in Francis's use of his 2015 encyclical—a letter on important matters to the Bishops and the wider church—to emphasise the importance of a diversity of voices when considering socio-scientific issues like sustainability [13].

The language-games of STEM, though, provide little space for the kinds of socio-cultural curriculum that would appear most relevant to Francis's teaching. As Zeidler [27] has noted, STEM emphasises the 'null' curriculum and draws attention to a perceived skills shortage, often presented as a crisis. In doing so, it ignores opportunities for a holistic sociocultural model in which socio-scientific issues are embedded. Indeed, as Zheng [12] has argued, STEM actually continues the cold-war era history of normalizing a techno-

scientific response to a perceived world crisis. In doing so, STEM expertise is positioned as the moral and intellectual guard of a fused future/global society that has been taught how to feel about change in the post-colonial world.

If, as Wittgenstein suggests, language and practice are interwoven, then it is difficult to imagine how teachers might find a coherent practice that incorporates the very different and complex [28] games surrounding scientific and technical education at this time.

## 2. Materials and Methods

*Finding the 'Blueprint' for the Game*

Writing a critique of a policy agenda like STEM for the pages of a critical journal is one thing. Capturing the interactions and interweaving of these language-games within teacher practice is quite another. To attempt to do so we have turned to the methodology of phenomenography [29], an approach that is also known as or associated with variation theory [3]. This approach seeks to investigate variations in human understandings of reality. That is, it is interested in the variations in discernment and experience of a phenomenon [30], and the conceptions people hold about that phenomenon [30]. It has been used extensively in educational research to explore epistemically different ways of seeing the object of learning [31,32].

For the analysis reported below, we used phenomenography to analyse a set of interviews collected with teachers within a large STEM curriculum project. The project was the major response by the Catholic school system in the state of South Australia to an array of STEM education policies from both state and national governments seeking to increase and improve STEM education in all Australian schools [33,34]. In total, 39 schools, 66 teachers and 841 students were involved in various aspects of the project which primarily involved teacher-led inquiry and collaborative evaluation.

The authors of this paper, STEM experts from the local university and the school system's central office, worked as consultants providing advice on both pedagogical and evaluation options. Strict protocols were in place, though, to ensure that teachers led the direction of the project in their school and their classroom. Participation in the research collection leading to this paper was opt-in and many teachers participating in the curriculum project chose not to participate in this research. The project was available by expression of interest (EOI) to all metropolitan schools in the school systems. Both primary and secondary schools participated and received limited financial support from the system to do so.

All school staff who participated in the project were invited to be part of the interview collection from which the data for this paper is drawn. The protocol involved the use of one-on-one semi-structured interviews conducted with both principals and teachers. To further minimise the risk of any kind of conflict of interest, the interviews were undertaken by a research assistant who had no other role within the project or the participating schools. The interviews were around 1 h in length, were conducted either in person or on zoom, and were undertaken with ethics approval from the human research ethics committees of both the University of South Australia and the school system. The audio recording of each interview was transcribed by the research assistant and participants were invited to review the transcript of their interview for accuracy and correction. In all, twenty school staff agreed to be interviewed, and the interviews with the 14 classroom teacher participants were used for the analysis reported in this paper.

Transcripts of the interviews were analysed using a computer-assisted technique supported by the *Leximancer* software [35]. An example of the new methodologies emerging in the digital humanities and social sciences, the method used has been set out in more detail elsewhere [2,36]. Briefly though, the method takes a corpus linguistic approach [37] and 'maps' the connections between concepts used across an entire sample of text—in this case the combined interview transcriptions. As with manual content analysis, the software looks for themes within the corpus by identifying how closely different concepts are used in relation to each other. The assumption is that if two concepts are consistently mentioned closely to each other—within a sentence or two—then those two concepts are related to

each other in people's thinking and communication [38]. The software does not tell us what the relationship between the two concepts means, just that it is present. The human researcher can then dig into the data to consider meaning as we might in a more traditional phenomenological study. For those not familiar with it, this methodology is most easily described in context, so we will explore it further along with the findings in the next section.

### 3. Results

Along with a full statistical account, the *Leximancer* software represents the connections between concepts in the form of an 'epistemic' or 'concept' map, as shown in Figure 1. To produce this map, the software identifies the most commonly connected words within the text. This varies from the more commonly encountered 'word clouds' which generally represent the words most frequently used within a text. In Figure 1 we can see the that the most commonly connected terms found across the combined interview data were 'able', 'students', 'STEM', 'year' and 'kids', and has positioned these as 'themes'.

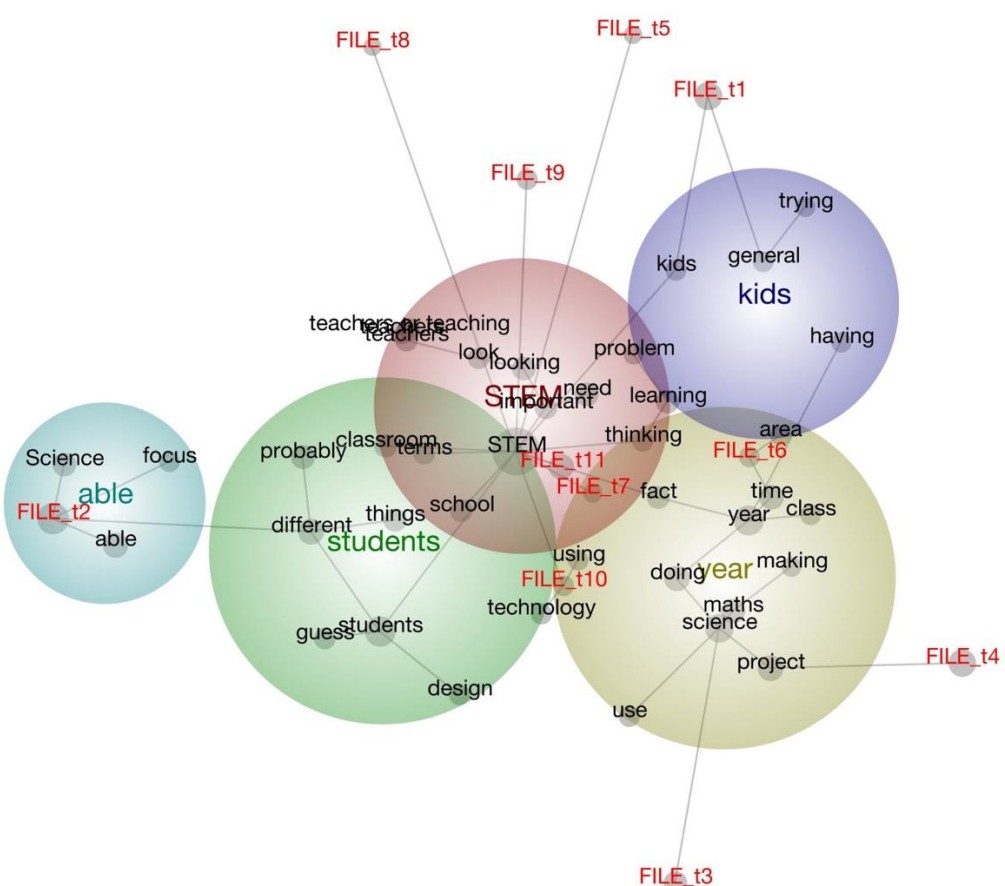

**Figure 1.** A map of the language games.

As already noted, the software is not identifying meaning in the text. It simply counts the distance between different concepts and represents the associations. In Figure 1 for instance, we can see that our interview group frequently use concepts like 'probably' and 'guess' in association with the concept of 'students'. The meaning for this example can be read relatively easily from the map. The teachers are being cautious in their identification of student thought or feelings, and a quick read of the underlying text confirms this. We can similarly see that 'STEM' seems to be a particular discursive choice as compared to 'science' and 'maths', with the later terms located together in a different part of the map to 'STEM'. The reason for this difference is less clear.

The software also supports a deeper reading of the underlying text by providing a printout of every example of the context in which a specified term was used. Guided by the

automated identification of connections within the interviews, this printout was analysed for meaning using the constant comparative method common in qualitative analysis [39]. The quotes used in the findings that follow are drawn from this contextual printout and are provided to aid trustworthiness in our analysis [39,40]. Through this deeper textual engagement, we are able to analyse for meaning. For example, we found a tendency for teachers to use to use the term 'STEM' when talking about high-level objectives but to use 'science' and 'maths' when talking about implementation. This clear discursive switch made when discussing different aspects of professional practice, this variation, is a clear example of Wittgenstein's language-games—language and practice have been interwoven to the point that different parts of practice have a different lexicon.

Our analysis worked this way throughout. We moved from the automated identification of associations down into a manual reading for meaning and back again. The automation did not provide us with interpretations, but it did allow us to focus in on our research interest quickly. It identified important variations in the language use that we are then able to interpret so as to identify the language-game.

Being semi-structured, clearly the interview schedule had some influence on what was discussed. The interview schedule, though, was wide ranging and provided multiple opportunities for teachers to talk about what they thought about the concept of STEM, what they were doing in the project, why they were doing it, and how what they were doing fit within a vision of Catholic education. To further check the interview schedule's influence, though, we were able to use a function of the software that positions each individual interviewee within the map of the entirety of the corpus. This appears in Figure 1 as a 'tag' in the format of 'FILE_tx' with 'x' being the unique identifying number assigned to each teacher. The fact that different teachers are positioned quite differently in the overall corpus—t2, for instance, spoke an entirely different language-game to t4—provides a strong indication that the interview schedule did not supress individual teacher thought or expression.

The tagging of individual contributions within the corpus is also very useful for the phenomenographic method, which is interested in variations in discernment, experience and understanding. Placing the individuals within the map allows us to quickly identify where variation exists. To see that there is significant variation between t4 and t2, for instance, but also that t4 and t6 have more nuanced differences in their reflections on the project. As we have noted already, however, the software is not able to tell us what those differences mean. Assigning meaning to the variation in language-games remains a manual task—one that we pursue in the next section of this paper.

*The Architectures of Variation*

An inspection of the concept map in Figure 1 reveals a distinct split in the discussion taking place in the interviews. The concept 'STEM' is central, but the concepts to the right and above that central concept are discussed quite separately to the concepts to the left of the map. That is, there is clear variations in the language-games or our interviewees. This occurs both because some teachers are adopting a discourse that is distinctly different to that of their colleagues, and because all of the teachers talk about different clusters of ideas in different parts of their interviews.

Some of this variation is meaningless. An idiosyncratic choice of words, nothing more. Sometimes, however, the discursive choices are layered with meaning. The apparently very similar concepts of 'kids' and 'students' is an example of this. This could easily be a simple case of individual habit—you say 'kids', I say 'students', but we both mean the same thing. When we unpacked the interview content, however, this was not the case. The teachers, we found, were making a distinct discursive choice and using 'kids' to signify something quite different to 'students'. This kids/students variation will be central to our argument below. First though, we will unpack the concept map as a whole to provide a sense of the wider 'game'.

The map in Figure 1 shows two levels of connection between concepts. The web of lines connecting the concepts shows a one-to-one relationship, the concepts are commonly found in the same sentence. The large circles, on the other hand, indicate broader themes within the text and show where statistically common clusters of concepts occur. The title for each theme is suggested by the software and is generally the most common or connected concept in the theme.

The centrality of the 'STEM' theme is to be expected, the teachers are being asked about their experiences within a STEM curriculum project. One thing evident in this theme is that the teachers involved tend to express a personal commitment to STEM. STEM is 'important', or a 'need'. The concept of 'thinking' is located in this theme and is linked very strongly to ideas of change. STEM is clearly seen as an opportunity to do things differently in order to increase student engagement. Examples of this conceptual use from the underlying text include:

> *And I think it's really understanding that engineering and technology can be the pencils use to write and the chairs that we sit on. We don't necessarily have to be thinking virtual reality or anything like it. It's about changing the way we think about those terms [engineering and technology] (t1).*

> *The 'music' came about [the teacher was saying that the plan came together harmoniously] with the designing of a powerful STEM enquiry—and the approach with empathy—and thinking of students in the class that don't engage well. So, that student in my class who I was thinking of who is quite bright but doesn't fulfil their potential in school (t10).*

The discussions leading to the theme 'year' were largely about the organisation of class activity. The word 'year' is used in phrases such as 'this year we are going to … ' or, 'with the Year 8 students we are … '. In either use, the teachers are talking about the changes they are making to teaching and learning within their classroom in very practical and descriptive ways. An interesting discussion that emerges in the 'year' theme, though, is the perceived challenge of 'matching' the maths and science content. This actually drives the presence of the 'maths' and 'science' concepts within the theme. The teachers are aware that the Australian curriculum is organised on the basis of year-level progression and that this may present challenges for implementing the project-driven learning they imagine STEM to be about. For instance:

> *Yeah 'cos I mean you can do STEM and you're using year 3 maths, or you're in a year 10 class, so it's about making sure that, okay well if we're actually going to do something, what kind of content descriptors do we actually want to hit here (t10).*

It is the remaining themes, though, that are the most interesting because of the discursive choices we have already identified. In the Australian idiom 'kids' is a word used to refer to children very informally. Teacher use of the word typically implies a level of parent-like affection for the children. In these interviews the teachers are choosing this informal use when they are talking about the mechanics of the classroom and the learning activities. More than this, the teachers are using the term when they are talking about what they are doing or designing *for* the kids. The discussion generating this theme takes place from a largely teacher-centric standpoint, for example:

> *Where the STEM comes into it, where they have got their devices floating they need to get it to float at a particular angle in water so that it will be calibrated for different densities and then they will do a little science experiment hopefully first week back–this has been dragging on. Then they will do a bit of a science … experiment and then mathematically model the angle of tilt to a specific gravity so there will be some mathematical modelling so I wanted to bring in–a bit of everything to try and show kids oh well–in industry there is a thing called process control–control processes we need to monitor and these days we don't dip sticks into this that and the other (t7).*

The understanding signified by the choice of 'student', on the other hand, appears to be quite different. This choice appears related to ideas of what students are able to

do—and note that the theme 'able' is associated with the theme 'student'—through the project design. The discursive choice of student seems to signify the foregrounding of student agency and self-regulation. At the extreme end of this way of thinking is one teacher noting that 'we support our students in some ways, almost to their detriment when they leave us' (t1). This teacher is arguing that students need a greater capacity to look after their own learning. The sense of student agency is similarly present when thinking about the school grading game. Note here the switch of agency with the students taking responsibility for demonstrating their capacity:

> So, for me, I feel like my grades–it's not necessarily about the mean grade going up, but I feel that there are more opportunities for all students to be successful and demonstrate their understanding of at least the 'C' standard in one form or another. And that it's more purposeful because we're linking it to what we're covering at the time (t1).

Connected to this concept of student ability and self-regulation there is also some questioning of the object of learning.

> So that's been good . . . we went to a PD the week before where we spoke with 2 ladies from New Zealand just to get some ideas and one thing they spoke about quite a bit was the general capabilities over there. And how they, they do the backward design and they actually start off with, they, they start off with talking about their dreamtime stories and, and then the curriculum follows. So I think that connection has come through really great. And–and not so theological for the students to not understand it (t2).

The evocation of 'dreamtime stories' here speaks to a knowledge formation quite removed from STEM and its future-global assumptions. 'Dreamtime' or 'Dreaming' stories are unique stories and beliefs owned by different Australian Aboriginal groups. They are the lore of the people and can be seen as a foundation to religion and law. It is unlikely that the colleagues from New Zealand would be starting with stories of the Dreaming but rather from the traditional knowledge of the Maori. Nevertheless, in passages like this we see that the difference in the kids/students conceptualisation is more than a difference in teacher or student standpoint. Within the student conceptualisation we see a greater openness to students having agency not only as managers of their own learning, but also as participants in choosing the forms of knowledge construction with which they will engage.

Striking in this content analysis is that most of the teachers are deploying both the 'kids' and the 'students' formulations. This is not an idiosyncratic choice. They individually switch from one to the other. This can be seen in Figure 1 where all of the teachers apart from t2 are basically positioned 'between' the two themes. Only t2 seems to use 'students' alone, while t3 and t4 actually made limited use of either theme. The rest of our participants, though, used both language-games. Given that most of the teachers readily make use of both discursive arrangements, it is also striking that there are no cross links across the concept map other than through the centrality of 'STEM'. This is because the switch of discursive choice happens entirely when the activity being discussed also switches.

## 4. Discussion

Our interest in this paper was to explore the impact of the tensions we saw between the language-games of the STEM policy agenda and other agendas within Catholic education on teacher practice. The data we interrogated, a series of interviews that were essentially an oral history of a large curriculum project, did not generate an explicit answer to our research question. Nobody said, 'the STEM agenda made me do this' or, 'the Papal encyclical made me do that'. Perhaps with one or two exceptions, the teachers did not have a well-developed abstract conception of any of these agendas. If anything, they used the interview with our researcher assistant, who was not an expert on these agendas, to seek some clarifications.

Never-the-less we would argue that through the choice of language-games they revealed, the interviews did tell us a great deal about teacher response to tensions in the cultural-discursive arrangements [1] forming a 'site' for this project. To make this argument, we will borrow the concept of 'teacher democratic assignment' from the recent work of

Mooney Simmie and Edling [4], building on the work of Englund [41]. This work suggests four 'democratic assignments'—similar to what Kemmis describes as cultural-discursive arrangements—for teacher professional practice that they name perennialism, essentialism, progressivism and reconstructivism. Our argument is that these 'assignments' work well as a summary of the tensions we've been exploring in this paper.

Each 'democratic assignment' offers a quite different epistemic positioning, a different 'architecture', for the work of teachers within wider society. Distilling the spirit of the enlightenment, for instance, essentialism valorises scientific rationality and holds that the curriculum should be grounded in empirical knowledge. In the western tradition essentialism separates from perennialism, which has a greater focus on traditional Christian, humanistic and, at times, nationalistic ideologies, often presented as the transmission of the 'western canon'. Sharing much with progressivism, on the other hand, reconstructivism understands democracy as a dynamic system of discursive space, context, interactions and interpretations. While still greatly concerned with disciplinary knowledge, both progressivism and reconstructivism draw our attention to the moral consequences of thoughts and actions in everyday life [42]. Reconstructivism, though, understands the concern of education as moving beyond the individual and as having an immediate role in creating the affordances for social and political change.

With this democratic assignment model in mind, we see great similarities in the architectures of both the STEM agenda and the 'kids' theme that emerged in the interviews. Both call heavily on an essentialist cultural-discursive arrangement. Both offer very narrow conceptions of curriculum and of democracy. Both position content as neutral rather than value laden, as Zeidler [27] has shown how this is explicit in science curriculum documents. And both position teachers as technicians and transmitters of knowledge with a role in securing social order [43].

With its default pedagogy of student-led inquiry and regular emphasis on critical and creative thinking, STEM does briefly challenge the concept of 'teacher as transmitter'. The STEM narrative, however, relentlessly assures students that the skills and knowledge of STEM are essential for the future [33] and are otherwise value free. The pedagogies may be constructivist, but students are expected to obediently construct knowledge and new practice within approved epistemologies that are strongly aligned with labour market needs.

The other language-games we have seen in this project, however, have different alignments. The 'students' theme in our interviews reflect a progressive discursive arrangement and seeks to support students to thrive in ways that reach well beyond the needs of the labour market. The objective of education here is tied up with ideas of student self-regulation and self-actualisation. The discourse we see captured in the Papal encyclical and the *Living Learning Leading* curriculum framework, on the other hand, suggests a reconstructivist arrangement. Far more than the teacher discourse, this policy agenda of Catholic education and the Catholic Church seeks to promote social, if not political change. It seeks not only to prepare young people for a good life within the world, but to also change the world in ways that promote what the church sees as a good life. It asks young people to be not only reflective, but also reflexive.

In this lost summer in Australia the differences between essentialism and reconstructivism seemed so stark. Although lost in the pandemic that followed, as we lived through the fires the debate on what they meant and how we should collectively respond to them became almost as fierce as the fires themselves. Through it all, though, our Government held to an essentialist line that 'any action should not put at risk a single job' [44].

Analysing, or even recounting, the power of vested interests in this debate is beyond the scope of this paper. As a sketch of our world, though, this debate is instructive. We see here a Government offering an essentialist position in which science and technology are subordinated to a pre-imagined future in which the nature of human labour has been determined. The loss of animal and human life, property and habitat is tragic and 'thoughts and prayers' are offered. Our focus, though, so we were told, must remain on being competitive within the current global economic system. There was—and remains—a

resistance to the call from so many in the community to reconsider if this is the world worth living in. To ask in the language of the Living Learning Leading framework if this is the world God desires.

The tension between the essentialist and reconstructive cultural-discursive arrangements, however, may not be the most important for teacher professional practice. What we have seen in the analysis outlined in this paper is that teachers seem able to connect, to interweave, the language and the practice of the essentialist position. On the other hand, while they seem readily able to articulate and implement a progressive practice, they do not appear to have a practice response to the quite reconstructive frameworks of their Catholic system when thinking about education in the scientific and technical subjects covered by STEM. They articulate a wider vision for the needs of their students than the 'jobs of the future' mantra of the STEM agenda, but they do not express any clear vision of how their work will support the realisation of a better world.

It is this tension between the progressive and reconstructivist cultural-discursive arrangements that seems to most limit the implementation of curriculum innovation in line with the frameworks of the system when compared to teacher capacity to implement the reforms called for by the STEM agenda. With this in mind, we will conclude this paper by outlining the case for more work being done on understanding STEM practices in light of the reconstructivist arrangements we find in frameworks such as Living Leading Learning.

## 5. Conclusions

The tensions we have discussed in this paper go out beyond the confines of education. Scholars of the sociology of science, for example, have shown the increasing challenges of achieving a productive engagement with scientific expertise when engaged in socio-technical decision making. As we have done in this paper, this case is often made calling on the example of climate change [45] and it explores the need to reimagine how society interacts with the particular kinds of expertise offered by the sciences [15,46]. The need for reimagination, it is argued, is being driven by a wider break down in the architectures that underpinned a social 'faith' in science from the enlightenment until the late twentieth century [21,47].

The research reported in this paper suggests a similar need for reimagining practice within education. This research suggests that in the context of contested and rapidly shifting practice architectures, the practices that have emerged are not yet coherent. Understandably, teachers are responding to competing demands by enacting multiple and quite separate practice logics, making it difficult realize a project that 'hangs together'. The impact of this is evident in the ongoing challenge of sustaining curriculum innovation under the STEM project [48]. As we have seen in the research reported in this paper, teachers do not appear to have a language to resolve the conflicts of purpose presented by the architectures shaping their practice.

The argument we have been making throughout this paper is that the need for teachers to be able to develop a more coherent set of, and sense of, STEM practices [5] is pressing. In a world dealing with increasingly complex and even existential challenges, how teachers and how wider society make sense of scientific expertise really matters. To fully engage in the task of sense making, though, teachers will need support in developing the skills needed to engage in the language games they encounter in the cultural-discursive arrangements that shape their practice. They will need a language that will allow them to connect the different discursive arms we see in Figure 1 and develop a clear and coherent response in their practice. And here lies a challenge for future professional learning. While developing the skills of new pedagogy is an essential ambition, it must be matched with the development of capacity to engage in the language-games of the cultural-discursive arrangements of teacher practice.

Education is implicated here. The danger of another generation of uninvolved, unengaged and uninformed citizens is becoming existential.

**Author Contributions:** Conceptualization, S.N.L., L.O. and B.W.; methodology, S.N.L.; formal analysis, S.N.L., L.O. and B.W.; investigation, L.O. and B.W.; resources, M.O. and K.S.; data curation, L.O.; writing—original draft preparation, S.N.L.; writing—review and editing, S.N.L.; visualization, S.N.L.; project administration, B.W., K.S. and M.O. All authors have read and agreed to the published version of the manuscript.

**Funding:** This project was supported by funding from Catholic Education South Australia.

**Institutional Review Board Statement:** The study was conducted in accordance with the Declaration of Helsinki, and approved by the Human Research Ethics Committee of the University of South Australia (Approval Number 200901) for studies involving humans.

**Informed Consent Statement:** Informed consent was obtained from all subjects involved in the study.

**Data Availability Statement:** The data that support the findings of this study are available from the corresponding author, SL, upon reasonable request.

**Conflicts of Interest:** All authors were involved in the curriculum project that this paper is drawn from. Their role was that of consultants supporting the schools and teachers involved to explore new curriculum and pedagogical approaches. The research on teacher learning from this position was transparent. Two authors, M.O. and K.S were employees of the funding organisation at the time of writing. The final version of this paper was approved by the funder.

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
