# Peer review of "The STEM Crisis and Teacher Practice: Exploring Responses to the Competing Discursive Arrangements of Education in the Sciences in a Catholic School Setting"

_education, doi:10.3390/educsci12100709_

Round 1
Reviewer 1 Report
This is an interesting paper on an important topic. The call to alarm in the introduction and conclusion are interesting and the analysis provided seems credible. The paper does not read like a typical peer-reviewed journal article with the personal experience and existential crisis that begin and end the paper. If this is appropriate for Education Sciences, then I found them interesting and set the tone for the paper. The authors extrapolate a great deal from Figure 1. This is not my area of specialization or research, so I am not an excellent judge of the quality of the research. The findings are clear and interesting and make for a provocative read.
Author Response
Dear Maria,
Please find uploaded a revised version of manuscript education-1884265, The STEM crisis and teacher practice. As always, we thank all three reviewers for their time and their consideration of our research.
As you are aware, reviewers 1 and 2 provided quite consistent and positive feedback. Reviewer 1 concluded that this ‘is an interesting paper on an important topic’, that the ‘findings are clear and interesting and make for a provocative read,’ and that the ‘call to alarm’ at the start of the paper was interesting. Reviewer 1 did express some minor uncertainty about the presentation of empirical results but acknowledged that was due to their own lack of familiarity with the methodology. As described below, we have sought to provide a stronger explanation of the research methodology in this version of the paper.
Reviewer 2 also had a positive reading of our paper, suggesting only that the project be repeated in a different, non-Catholic, context in the future. We agree that to do so would be interesting, and we will do so when the opportunity presents itself. In this revision we have also sought to clarify that the alternative ‘language’ games provided through Catholic thought can be found in diverse places – we’ve cited the OECD and John Dewey as just two examples.
Reviewer 3, in contrast, has suggested extensive revisions. Upon engaging with reviewer 3’s feedback, we have concluded that they have somewhat misunderstood the intent of the paper. We acknowledge that the structure of the original paper was likely the cause of any confusion and have subsequently made the major revisions attached.
We note particularly that Reviewer 3 understood the purpose of the paper to be ‘an exploration of the experiences of teachers involved in a STEM curriculum project’, which is actually a distinctly different theoretical and methodological research focus to our stated research question of ‘how do the tensions in the language-games of policy impact on teachers’ practice in this STEM curriculum project’? Our major revisions, therefore, have focussed on making our core argument easier to follow. We have done this through a major re-writing of the introduction, materials and methods, and results sections of the paper. We will never-the-less respond to the specific items raised by reviewer 3 below:
- It is unclear how the preface links to the study on teacher experiences of a STEM project. It is noted that there is an explanation that climate change is "failures of our socio-scientific decision making" but these present as isolated statements when the policies and strategies along with background literature is not more comprehensively reported. The discussion section where it is raised again presents as being disjointed to the results section. The preface is written from a first person perspective when there are multiple authors which contributes to the previous point about its purpose being unclear. It is recommended that the inclusion of "climate change" is reviewed.
We have repositioned the ‘architecture for this research’ section to later in the introduction (now at line 89) and included more detail to explain its presence (line 90-105).
- Given that the manuscript reports on an exploration of the experiences of teachers involved in a STEM curriculum project, there is limited background literature reviewed in the introduction. It is recommend that this is addressed with additional literature included in this section.
The purpose of the paper is actually to better understand the ‘language games’ that formed the architectures for teacher practice within the project. We have sought to bring greater clarity to this purpose by adding a new section to the start of the paper that is less ambivalent than our previous ‘call to alarm’ without context (lines 20-64).
We have additionally moved the literature review on both STEM as policy (lines 67-87) and, more importantly, on language-games (lines 135-214) into the introduction. We have added to both.
- The manuscript details "The project we are exploring here was the major response by this large Catholic school system to an array of STEM education policies and strategies at the state and national level". It is unclear if the examined STEM curriculum project aims to address these policies and strategies or specific policies and strategies. It is recommended this is clarified and examples of these policies and strategies are included.
We have responded to this suggestion through additional citation. As our purpose is not to explore the teaching strategies per se, but rather teacher reactions to their purpose, we think it is best not to add to what is already a quite lengthy paper.
- Direct quotes require additional information in the in text reference. For example: ‘dialogue with the world as a people of faith’ and sets out the primary purpose of schooling as the development of ‘thriving people, capable learners, leaders for the world God desires.’ and ‘between 2006 and 2011 in Australia, the number of people in positions requiring STEM qualifications grew 1.5 times faster than all other occupations groups’.
This has been addressed.
- The materials and methods section does not include important information to understand the research approach. This includes:
- State/territory location in Australia given an array of STEM education policies and strategies at the state is being explored.
- School type
- How were the schools selected
- Recruitment of schools and participants
- Participants: Number, number at each school and relevant information to support the research approach
- Information relating to the interview process: Individual/group, length, face to face/online, audio/video recorded
- Ethics and approvals to conduct the research statements
This information has been added at lines 228-239.
- There is text overlap at points in figure 1. Recommended that this is addressed so all text is clear readable.
This is a formatting error that is not present on our version. It will be picked up in the galleys.
- It is recommended that the results section commence with an overview of how the themes of "STEM", "kids", "students", "year" and "able" were identified and the definitions of these.
We have addressed this at lines 255-262, while also taking the opportunity to provide more detail on the methodology in response to the comments from reviewer 1.
- Whilst the selected direct quotes in the results section align to the themes, it is recommended that the quotes are reviewed or additional quotes included which relate to research topic - teacher experiences of the STEM curriculum project in a Catholic education system and school.
We again highlight that our interest was not in the teacher experiences per se, but in their entanglement in the language games that formed the practice architecture for the project. We have sought to step out the process of analysis for this purpose with greater clarity through adding lines 272-278, along with additional reference to related work.
We would argue that with this important clarification, the presentations of the data through quotes is sufficient to provide trust in our analysis. We feel that reviewers 1 and 2 support this position.
- In the discussion section it states "The interviews, though, did show us that directly asking the teachers about such high-level policy direction would likely have been a fruitless exercise". It is unclear where this is discussed in the results section and as such is recommended for inclusion.
This was something of a ‘throw away’ line and has been deleted.
Reviewer 2 Report
It is worth repeating the experiment in non-Catholic schools. Especially in which STEM is a daily practice. Perhaps there are no differences between Catholic and non-Catholic schools.
Author Response

(The authors gave the same response as above.)

Reviewer 3 Report
An interesting study that explores experiences of teachers involved in a STEM curriculum project.
The following are comments and suggestions for the authors:
1. It is unclear how the preface links to the study on teacher experiences of a STEM project. It is noted that there is an explanation that climate change is "failures of our socio-scientific decision making" but these present as isolated statements when the policies and strategies along with background literature is not more comprehensively reported. The discussion section where it is raised again presents as being disjointed to the results section. The preface is written from a first person perspective when there are multiple authors which contributes to the previous point about its purpose being unclear. It is recommended that the inclusion of "climate change" is reviewed.
2. Given that the manuscript reports on an exploration of the experiences of teachers involved in a STEM curriculum project, there is limited background literature reviewed in the introduction. It is recommend that this is addressed with additional literature included in this section.
3. The manuscript details "The project we are exploring here was the major response by this large Catholic school system to an array of STEM education policies and strategies at the state and national level". It is unclear if the examined STEM curriculum project aims to address these policies and strategies or specific policies and strategies. It is recommended this is clarified and examples of these policies and strategies are included.
4. Direct quotes require additional information in the in text reference. For example: ‘dialogue with the world as a people of faith’ and sets out the primary purpose of schooling as the development of ‘thriving people, capable learners, leaders for the world God desires.’ and ‘between 2006 and 2011 in Australia, the number of people in positions requiring STEM qualifications grew 1.5 times faster than all other occupations groups’.
5. The materials and methods section does not include important information to understand the research approach. This includes:
a. State/territory location in Australia given an array of STEM education policies and strategies at the state is being explored.
b. School type
c. How were the schools selected
d. Recruitment of schools and participants
e. Participants: Number, number at each school and relevant information to support the research approach
f. Information relating to the interview process: Individual/group, length, face to face/online, audio/video recorded
g. Ethics and approvals to conduct the research statements
6. There is text overlap at points in figure 1. Recommended that this is addressed so all text is clear readable.
7. It is recommended that the results section commence with an overview of how the themes of "STEM", "kids", "students", "year" and "able" were identified and the definitions of these.
8. Whilst the selected direct quotes in the results section align to the themes, it is recommended that the quotes are reviewed or additional quotes included which relate to research topic - teacher experiences of the STEM curriculum project in a Catholic education system and school.
9. In the discussion section it states "The interviews, though, did show us that directly asking the teachers about such high-level policy direction would likely have been a fruitless exercise". It is unclear where this is discussed in the results section and as such is recommended for inclusion.
Author Response

(The authors gave the same response as above.)

Round 2
Reviewer 3 Report
It is recommend that the manuscript be reviewed for edits. For example this sentence is repeated While 151 couched in the language of the Catholic church, the framework reflects a wider philosophical interest in the ‘good’ human life.
It is recommended that any potential conflict of interest is addressed in relation to "with the authors of this paper 232 among the consultants providing advice on different pedagogical options."
The following information from the previous review was not included in the changed manuscript. It is recommended that this information be included: How many participants? How many teachers and how many principals? How were the participants recruited and selected for interviews, the participants school type. Information relating to the interview process: individual/group, length, face to face/online, audio/video recorded.
In the results section, 'teachers' are referred to. As the interviews were conducted with principals and teachers, were the results only related to the teachers?
Author Response
It is recommend that the manuscript be reviewed for edits. For example this sentence is repeated While couched in the language of the Catholic church, the framework reflects a wider philosophical interest in the ‘good’ human life.
A further proof reading has been conducted.
It is recommended that any potential conflict of interest is addressed in relation to "with the authors of this paper among the consultants providing advice on different pedagogical options."
We have amended section 2.1 to address this matter, particularly on line 233 and from line 240.
The following information from the previous review was not included in the changed manuscript. It is recommended that this information be included: How many participants? How many teachers and how many principals? How were the participants recruited and selected for interviews, the participants school type. Information relating to the interview process: individual/group, length, face to face/online, audio/video recorded.
We have addressed all of these matters in section 2.1, particularly from line 240.
In the results section, 'teachers' are referred to. As the interviews were conducted with principals and teachers, were the results only related to the teachers?
This is also addressed in section 2.1. The data used in this paper was only that from the teachers.